# MACKO: Sparse matrix-vector multiplication for low sparsity

**Vladimír Macko** [1 2]  **Vladimír Boža** [1]

## Abstract

Sparse Matrix-Vector Multiplication (SpMV) is a fundamental operation in the inference of sparse Large Language Models (LLMs). Because existing SpMV methods perform poorly under the low, unstructured sparsity $(30-90\%)$ commonly observed in pruned LLMs, unstructured pruning provides only limited memory reduction and speedup. We propose **MACKO-SpMV**, a GPU-optimized format and kernel co-designed to reduce storage overhead while remaining compatible with the GPU's execution model. This enables efficient SpMV for unstructured sparsity without specialized hardware units or precomputation. We identify memory bandwidth as the primary limiting factor of SpMV and analyze the storage overhead of MACKO. At $50\%$ sparsity, MACKO is the first approach to achieve $1.5\times$ memory reduction and $1.2$–$1.5\times$ speedup over the dense baseline as well as substantial improvements over other SpMV methods: cuSPARSE $(2.8$–$13.0\times)$, Sputnik $(1.9$–$2.6\times)$, and DASP $(2.2$–$2.5\times)$. An LLM pruned with Wanda to sparsity $50\%$ requires $1.5\times$ less memory and achieves $1.5\times$ faster inference at fp16 precision. As a result, **unstructured pruning at $50\%$ sparsity becomes practical** for real-world LLM workloads and **bridges the efficiency gap with structured 2:4 sparsity**.

## 1. Introduction

Large Language Models (LLMs) (Zhang et al., 2022; Touvron et al., 2023; Dubey et al., 2024; Team et al., 2024; Jiang et al., 2024; Abdin et al., 2024) have revolutionized natural language processing and related fields. Although they are typically deployed in large data-centers, there is a growing demand for deployments with limited number of GPUs and local consumer-grade GPUs.

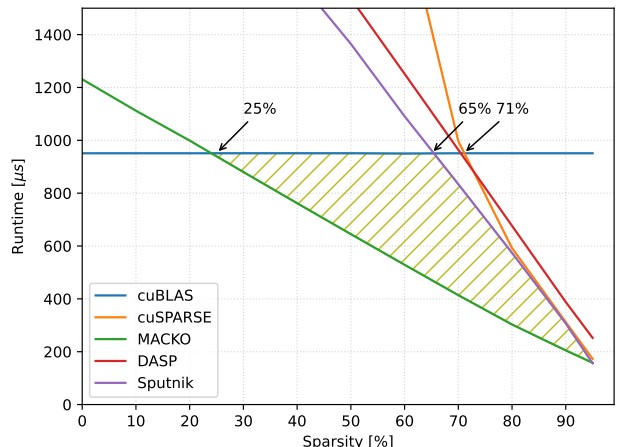

*Figure 1.* Sparse matrix–vector multiplication runtime. Matrix size 36864, 12288 in fp16 on an NVIDIA GeForce RTX 4090 GPU. Using MACKO, sparse computation exceeds the performance of dense at sparsity as low as 25%. Existing libraries require 2.6× fewer non-zeros to achieve the same performance. The improvement of this work is displayed as highlighted region.

Local inference faces major challenges due to limited hardware resources, especially memory capacity. Running the Llama2-13B model (Touvron et al., 2023), which requires about 26 GB of memory in FP16 precision, on a high-end consumer GPU such as the RTX 4090 with only 24 GB, requires compression techniques to reduce the model size and computational cost. Unstructured neural network pruning is one such promising method, allowing a large portion of weights to be removed (set to 0) with minimal degradation in model quality (Frantar & Alistarh, 2023).

The main bottleneck lies in the inefficient Sparse Matrix–Vector Multiplication (SpMV), the dominant computational kernel in local inference with sparse LLMs. SpMV is heavily used across the core layers, including QKV projections, up and down projections, and fully connected layers (Xia et al., 2023), with a typical sparsity between 50% and 90% (Lee et al., 2025). Much research has focused on improving SpMV (Greathouse & Daga, 2014; Kreutzer et al., 2014; Yan et al., 2014; Liu & Vinter, 2015; Merrill & Garland, 2016; Gale et al., 2020; Niu et al., 2021; Gómez et al., 2021; Zheng et al., 2022; Du et al., 2022; Lu & Liu, 2023; Lin et al., 2025), but all these methods suffer from one or more of the following disadvantages: they depend on

---
[1]Faculty of Mathematics, Physics and Informatics Comenius University Bratislava [2]GrizzlyTech. Correspondence to: Vladimir Macko <vladimir.macko@fmph.uniba.sk>.

*Proceedings of the $43^{rd}$ International Conference on Machine Learning*, Seoul, South Korea. PMLR 306, 2026. Copyright 2026 by the author(s).

special hardware support, require extensive precomputation, or are targeted at high sparsity levels.

To address these challenges, we propose **MACKO-SpMV**, the **M**utually **A**ligned **C**ompressed coordinates **K**ernel **O**ptimized for **Sp**arse **M**atrix **V**ector multiplication. MACKO introduces a novel format for representing unstructured sparse matrices based on coordinate compression with mutual alignment between coordinates and values. To achieve mutual alignment, MACKO employs strategic padding of sparse matrix elements. We show that this padding overhead is negligible in average cases and remains small and bounded even in the worst case. This enables efficient GPU implementation and as a result, state-of-the art memory reduction and speedup for $30 - 90\%$ sparsity.

We evaluate MACKO against three state-of-the-art SpMV implementations—`cuSPARSE`, `DASP` (Lu & Liu, 2023), and `Sputnik` (Gale et al., 2020), as well as a dense baseline `cuBLAS`, using different consumer GPUs. For FP16 precision at $50\%$ sparsity, MACKO is the first method that achieves a meaningful $1.2$–$1.5 \times$ speedup and $1.5 \times$ memory reduction over `cuBLAS`[1]. Furthermore, MACKO outperforms all baselines on all sparsity levels from $30\%$ to $90\%$ on all tested GPUs. These improvements directly lead to faster and more memory-efficient end-to-end sparse LLM inference.

Our key contributions are summarized as follows.

- We introduce the MACKO storage format, optimized for sparse matrices with sparsity 30–90%, and analyze its storage overhead.
- We design and implement an optimized SpMV GPU kernel and integrate it into the ML framework `PyTorch`.
- We conduct extensive experiments on three consumer GPUs, demonstrating that MACKO significantly outperforms existing SpMV implementations in both speed and memory efficiency.
- We show how MACKO bridges the gap between unstructured and 2:4 sparsity.

## 2. Background and Related Work

This work is primarily targeted at LLM pruning, but is applicable to all models with linear layers. See Section B for a brief overview of LLM inference.

### 2.1. NVIDIA GPUs

NVIDIA GPUs consist of multiple *streaming multiprocessors* (SMs) and a hierarchical memory structure. The smallest unit of execution is a *thread*. The threads are grouped into *thread blocks* that form the *grid*. 32 threads within a

block are grouped into a *warp* and are executed simultaneously in *single instruction multiple threads* (SIMT) mode on SM. The index of the thread within a warp is called *lane*.

The memory hierarchy consists of high latency *global memory* accessible by all threads, *shared memory* within each SM shared by threads in a thread block, and a fast but limited number of *registers* private to each thread (Guide, 2013). Access to global memory has approximately 15 times higher latency than access to shared memory, which in turn has approximately 30 times higher latency than access to registers (Luo et al., 2024).

Threads can also communicate with each other in a limited way with the use of warp shuffling. Based on the shuffling pattern, this communication can be as fast as register access or as slow as shared memory access.

### 2.2. Pruning

Pruning emerged as a promising model compression technique for reducing the computational and memory demands of LLMs. It reduces the number of non-zero weights by removing the least important connections. In case the sparsity structure is unconstrained, multiple methods were able to prune a large percentage of weights without much quality degradation (LeCun et al., 1989; Frantar & Alistarh, 2023; Ma et al., 2023; Sun et al., 2023; Zhang et al., 2024; Boža, 2024; Boža & Macko, 2024; Lee et al., 2025). These methods mainly target sparsity around $50\%$ with recent work achieving sparsity up to $90\%$(Lee et al., 2025). Unfortunately, they often provide no or very limited practical memory reduction or speedups.

The main bottleneck is the lack of efficient formats for storing sparse matrices and fast SpMV GPU kernels. Many works proposed methods to improve SpMV through vertical slicing (Gómez et al., 2021; Kreutzer et al., 2014), 1D tiling (Gale et al., 2020), 2D-tiling (Yan et al., 2014; Niu et al., 2021), balancing workload by reconstructing nearly even-sized basic working units (Greathouse & Daga, 2014; Liu & Vinter, 2015; Merrill & Garland, 2016), utilization of hardware acceleration (Zheng et al., 2022; Lu & Liu, 2023), extensive precomputation and optimization (Du et al., 2022; Lin et al., 2025). All these approaches suffer from one or more of the following disadvantages: they depend on special hardware support, require extensive precomputation, enforce specific sparsity patterns or are targeted at high sparsity levels.

A common way to address the problems with sparsity is to impose constraints on the structure of non zero elements (Lagunas et al., 2021; Ma et al., 2023; Ashkboos et al., 2024). Different types of constraints were proposed and successfully utilized to achieve practical performance benefits. The most popular are: block sparsity which allows

---

[1]`cuBLAS` is the strongest baseline at this sparsity level.

sparsity only at a level of blocks (for example, the whole $8 \times 8$ must be filled with non zeros or be completely empty) (Borštnik et al., 2014), and N:M sparsity (usually 2:4) which decomposes the matrix into blocks with $M$ elements and enforces $N$ out of those elements to be non zero (Lin et al., 2023; Castro et al., 2023). However, enforcing structural constraints often leads to an inferior model accuracy and less flexibility in model design.

## 2.3. System level optimizations

In practice, multiple other factors influence the final performance of the LLM serving system. Optimization of these systems focuses on restructuring computation graphs with ByteTransformer (Zhai et al., 2023) and DeepSpeed (Aminabadi et al., 2022) or refining attention mechanisms with FlashAttention (Dao et al., 2022; Dao, 2023; Shah et al., 2024). Especially relevant in a memory constrained environment are offloading methods such as those implemented in FlexGen (Sheng et al., 2023) and llama.cpp (Gerganov, 2023), which optimize memory usage by distributing model components across various hardware resources. MACKO targets weight pruning, is independent of these methods, and can be combined with them to further enhance performance.

# 3. Gaps and Opportunities

## 3.1. Bottlenecks in SpMV

To analyze the bottlenecks in SpMV, we employ the Roofline model (Williams et al., 2009) and consider the Compute Intensity (CI) of dense matrix-vector multiplication (MV) and sparse matrix-vector multiplication (SpMV) for 16-bit values compared to ops per byte of modern GPUs.

Modern GPUs can generally perform much more operations than byte movements in a given time. This is captured by quantity *ops per byte* (OPB), defined as the ratio of peak floating point operations per second (FLOPS) to peak memory bandwidth (BW). For common GPUs OPB $\gg 1$ (80 for RTX 4090, 38 for RTX 3090, and 45 for RTX 2080).

For a matrix with $R$ rows and $C$ columns, the compute intensity is defined as the ratio of operations performed to the number of bytes read from and written to memory.

$$CI_{MV} = \frac{2 \cdot R \cdot C}{2 \left( R \cdot C + R + C \right)} \approx 1 \quad (1)$$

Given a matrix density $d = \frac{nnz}{R \cdot C} = 1 - sparsity$, we define *effective density effd*, a measure of how much memory the sparse matrix requires (*storage*) compared to dense representation, counting all bytes required to store the $b_{val}$-bit values and all bytes required to store the sparsity structure.

$$effd = \frac{storage}{R \cdot C \cdot b_{val}} \quad (2)$$

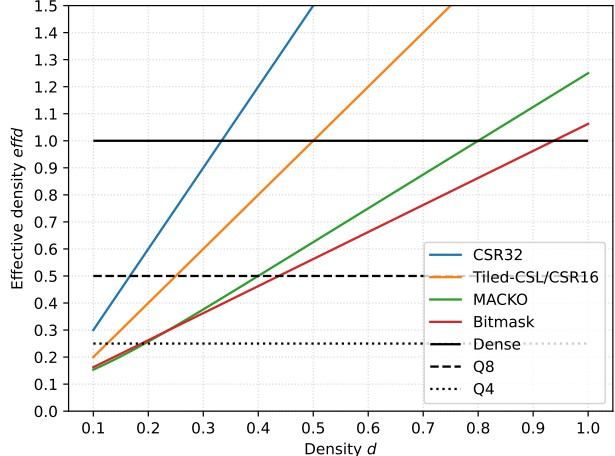

*Figure 2.* Effective density of different sparse matrix formats for 16-bit values. Q8 and Q4 reffer to quantization to 8 and 4 bits respectively. Expected effective density is shown for MACKO.

For sparse matrix-vector multiplication, the number of necessary operations decreases and the number of accessed bytes depends on the effective density *effd*.

$$CI_{SpMV} = \frac{2 \cdot d \cdot R \cdot C}{2 \left( effd \cdot R \cdot C + R + C \right)} \approx \frac{d}{effd} < 1 \quad (3)$$

Because $OPB \gg CI_{SpMV}$ and $OPB \gg CI_{MV}$, both MV and SpMV are primary limited by memory bandwidth. This is a great opportunity because improving the effective density of the sparse matrix format directly translates into less time spent on memory operations and improved performance of SpMV, as long as the computation stays compatible with the SIMT execution model and the computational overhead is sufficiently small (up to 38 operations per byte).

## 3.2. Analysis of storage formats

Improving the memory of the storage format directly translates into lowering the lower bound on SpMV execution time. We conduct a comparative analysis of several widely-used sparse matrix formats: CSR used by Sputnik (Gale et al., 2020) and other CUDA-core SpMM implementations, Tiled-CSL used in Flash-LLM (Xia et al., 2023), and bitmask (Fan et al., 2025). To make the equation simpler, we disregard terms that are negligible for large matrices, such as row pointers in CSR or constant terms.

In ideal case the format would only store non zero values and $effd = d$ which is not achievable in practice[2]. On the other hand, dense representation provides $effd = 1$.

**CSR32/CSR16** represents sparsity by storing column in-

---

[2]$effd = d$ is achievable only if the sparsity pattern is fully predetermined.

dices for each non-zero value as 32 or 16 bit integers, resulting in an effective density of:

$$effd_{CSR32} = d \cdot \frac{32 + b_{val}}{b_{val}} \quad (4)$$

$$effd_{CSR16} = d \cdot \frac{16 + b_{val}}{b_{val}} \quad (5)$$

**Tiled-CSL** divides the matrix into $NT$ tiles and stores column indices within each tile using 16 bits. For each tile, it also stores its starting offset using 32 bits.

$$effd_{Tiled-CSL} = d \cdot \frac{16 + b_{val}}{b_{val}} + \frac{32NT}{R \cdot C \cdot b_{val}} \quad (6)$$

The size of these tiles is usually set to at least $128 \times 64$, making the number of tiles $NT$ negligible compared to the total number of values $R \cdot C \cdot b_{val}$ for large matrices.

**2:4 sparsity** imposes a constraint on the sparsity structure that enforces at most 2 elements to be non-zero for every 4 elements. It represents the matrix as a list of non-zero elements with 2 bit index for each element and is only applicable at density 0.5.

$$effd_{2:4} = 0.5 + \frac{1}{b_{val}}. \quad (7)$$

**Bitmask** is one of the best formats in terms of effective density in a high density setting [3]. It uses one bit per value of the original matrix to indicate whether it is zero or nonzero and an array of non-zero values. The size of the bitmask is independent of the matrix sparsity and sparsity pattern, making it a robust choice for high-density matrices.

$$effd_{bitmask} = d + \frac{1}{b_{val}}. \quad (8)$$

Even though this format offers great compression and was successfully used on CPUs (Kurtic et al., 2025), it suffers from poor memory access patterns on GPUs. We are unaware of any efficient GPU SpMV implementation using bitmasking, although it has been used successfully in the matrix-matrix multiplication setting (Fan et al., 2025).

Figure 2 shows effective densities across density levels of the input matrix. MACKO has a lower effective density than CSR formats throughout the region of interest and even outperforms bitmask encoding at low density $d < 0.23$. Using MACKO to match the compression rate of 8-bit quantization, it is sufficient to achieve $60\%$ model sparsity compared to $83\%$ for CSR.

## 4. Design

The design of MACKO-SpMV is driven by two main goals: (1) minimize the effective density and (2) keep the SpMV algorithm compatible with the GPU parallelization model.

---

[3]Provably optimal for density $50\%$, see Section C.

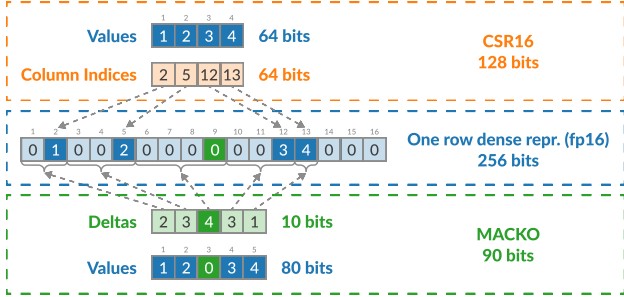

*Figure 3.* Example of MACKO storage format with 16 bit values and 2 bit deltas ($b_{val} = 16$ $b_\Delta = 2$) compared to CSR16.

### 4.1. Compressed storage format

MACKO is inspired by the well-known CSR format. Recall that CSR represents the sparse matrix $M$ as 3 arrays:

- `values`: list of non zero values of $M$ in row major order ($nnz$ entries, $b_{val}$ bits per entry).
- `column_indices`: column index of each value ($nnz$ entries, 16 or 32 bits per entry).
- `row_pointers`: indices of row beginnings in the `values` and `column_indices` arrays ($R + 1$ entries, 32 bits per entry).

MACKO takes the `column_indices` for each row and compresses them using delta encoding. Then it limits the deltas to a predefined range of $[1, 2^{b_\Delta}]$ by selectively including 0 values in the final encoding with a straightforward greedy strategy. Instead of storing these deltas, we store delta-1 as $b_\Delta$-bit unsigned integer.

Added padding increases the number of encoded values from the original matrix $M$ and the length of the arrays that hold values and deltas increases from $nnz$ to $pad\_nnz$. We show that this overhead is well bounded in the worst case and almost negligible in the expected case. Mutual alignment of compressed coordinates and values enables efficient GPU kernel design and fast execution.

The resulting MACKO format looks as follows:

- `values`: list of non zero values of $M$ in row major order with 0 for padding ($pad\_nnz$ entries, $b_{val}$ bits per entry).
- `deltas`: delta encoded `column_indices`, with fixed bit width ($pad\_nnz$ entries, $b_\Delta$ bits per entry).
- `row_pointers`: indices of row beginnings in the `values` and `column_indices` arrays ($R + 1$ entries, 32 bits per entry).

Figure 3 shows an example for $b_{val} = 16$ and $b_\Delta = 2$ for one row having values $[1, 2, 3, 4]$ with column indices $[2, 5, 12, 13]$ (1 based index for simplicity), MACKO inserts one 0 at columns 9. The values with padding will be $[1, 2, 0, 3, 4]$ and the deltas $[2, 3, 4, 3, 1]$. MACKO trades

a significant decrease in the size of deltas compared to the column_indices for a modest increase in the size of values.

Because GPU memory storage is organized into 8-bit words, we pack $\frac{8}{b_\Delta}$ values together using bit shifting. This limits the natural values for $b_\Delta \in \{1, 2, 4, 8\}$. Although MACKO can be used with many combinations of $b_{val}$ and $b_\Delta$, **we focus on $b_{val} = 16$ and $b_\Delta = 4$** because it has good trade-offs between compression and generality across all density levels relevant for neural network pruning.

## 4.2. Effective density analysis

MACKO introduces overhead in the form of value padding. We analyze the impact of this padding on the effective density $effd_{MACKO}$ under 3 scenarios: the best case, the expected case, and the worst case.

**Best case**. If the difference between all column indices is at most 16, no value padding needs to be introduced, and only overhead comes from the deltas.

$$best\_effd_{MACKO} = d\frac{b_{val} + b_\Delta}{b_{val}} \qquad (9)$$

**Worst case**. If all zeros in the original matrix $M$ are in segments of length 16, they all need to be covered by padding, and we need to add $R \cdot C \cdot \frac{(1-d)}{2^{b_\Delta}}$ new values to MACKO.

$$worst\_effd_{MACKO} = \left(d + \frac{1-d}{2^{b_\Delta}}\right)\frac{b_{val} + b_\Delta}{b_{val}} \qquad (10)$$

**Expected case**. We assume all elements of $M$ have a uniform, independent probability $d$ of being non zero and define $z = (1-d)^{2^{b_\Delta}}$. The expected density of the padding elements can be computed as a sum of geometric series and ends up being $d\frac{z}{1-z}$.

$$exp\_effd_{MACKO} = d\left(1 + \frac{z}{1-z}\right)\frac{b_{val} + b_\Delta}{b_{val}} \qquad (11)$$

$\frac{z}{1-z}$ starts to be noticeable only at low density $d < 20\%$, and is dominant at very low density of $d < 4\%$. The assumption that non-zero elements occur with uniform and independent probability may be too strong for practical use-cases. Sparsity patterns in LLM pruning could exhibit adversarial structure, however empirical results at Section 5.2 shows that to not be the case.

Thanks to this new format, **unstructured sparse matrices** match the memory footprint of dense representation for density as high as 0.78 (sparsity as low as 22%).

## 4.3. SplitK Matrix-Vector Multiplication

MACKO is based on a SplitK GPU algorithm for computing general matrix multiplication (GEMM) (Corporation, 2025),

adapted for general matrix-vector multiplication (GEMV). The work is naturally parallelized across rows of the input matrix $M$, with each row $M_r$ assigned to one warp of 32 threads. The threads in one warp collaborate to compute one value in the result vector $Y_r$ as a product of one row $M_r$ and vector $V$. The computation is parallelized across rows and across the common matrix dimension (in matrix-matrix multiplication commonly denoted as $k$).

For a given row $r$ at each step $i$, the threads in the warp collaboratively load 32 consecutive elements from $M_r$ and from $V$, one element per thread[4]. Then each thread $t_{lane}$ computes the corresponding product $M_{r,32i+lane}V_{32i+lane}$ and the computation advances to the next consecutive 32 elements. After the whole row is iterated through, each thread holds part of the final product. These parts are summed using warp shuffling primitives and stored in the result matrix.

## 4.4. MACKO-SpMV

MACKO parallelizes the work in the same way as SplitK algorithm. Each row $r$ of $M$ is processed independently by a warp of 32 threads that collaboratively compute one output $Y_r$. MACKO is parametrized by $b_{val}$, the number of bits required to store a nonzero value, and $b_\Delta$, the number of bits for each delta. We present the case where $b_{val} = 16$ and $b_\Delta = 4$.

MACKO computation can be split into the following parts: (1) **loading deltas and values from $M$** (2) **reconstructing column indices** (3) **loading values from $V$ and final computation**.

We describe the algorithm for $r$-th row $M_r$ of the input matrix $M$. The values and deltas for this row start at the index row_pointers[r] (inclusive) and end at the index row_pointers[r+1] (non inclusive).

### 4.4.1. LOADING DELTAS AND VALUES FROM $M$

At each step, MACKO loads and processes $load\_size = 8$ elements per thread. Each warp loads 128 consecutive bytes worth of deltas, which is four 8-bit words containing eight 4-bit deltas per thread. Each warp loads 512 consecutive bytes worth of values, which is eight 16-bit values per thread.

This is closely aligned with the hardware memory hierarchy and internal functions, because GPUs issue memory loads in 128 byte transactions. By selecting this load pattern, we are fully utilizing every memory transaction. We can make use of a similar memory loading pattern for other combinations of $b_{val}$ and $b_\Delta$, discussed in Section E.1.

---

[4]Each warp typically performs one memory transaction that loads 128 bytes, resulting in one 32-bit value per thread or two 16-bit values. For efficiency, a 512-byte transaction may be used, providing four 32-bit (eight 16-bit) values per thread.

### 4.4.2. RECONSTRUCTING COLUMN INDICES

Each thread holds $load\_size$ deltas in its registers. To reconstruct the column indices, each thread needs to know the sum of all deltas stored in the previous threads. The key insight of MACKO is that this sum can be efficiently computed without any impact on memory bandwidth, which only adds computational overhead, not a memory overhead. Because matrix-vector multiplication is a memory bound, this overhead **does not** translate to runtime slowdown.

To perform the sum, each thread first computes the sum of its $load\_size$ deltas ($local\_sum$) sequentially. The threads then collaborate to compute an exclusive prefix sum of the $local\_sum$ values across the warp. This is done using a binary reduction tree with depth $\log_2(warp\_size) = 5$. At each level $l$ of this tree, each thread holds the sum of previous $2^l$ threads. After Algorithm 1 is called, each thread will hold $prefix\_sum$ of all deltas held by the previous threads. Finally, all threads can sequentially compute column indices corresponding to their values loaded from $M$.

At the end of each step, the last thread knows the sum of all deltas used in that step. We broadcast this value to all other threads in constant time using `__shfl_sync(sum, 31)`.

### 4.4.3. LOADING VALUES FROM $V$ AND FINAL COMPUTATION

After each thread reconstructs the original column indices, it can load the corresponding value from the vector $V$ and multiply it with the value already loaded from $M$. Each thread performs this for all its $load\_size$ values and accumulates the sum of these products. After processing the whole row, each of the threads holds a partial dot product of the result $Y_r$. We sum these partial results again using warp shuffling and store the final result.

### 4.5. Optimization techniques

**Vectorized loading** is an important tool for mitigating bandwidth bottlenecks (Luitjens, 2013; Gale et al., 2020). It requires each thread in a warp to read consecutive chunks of 4, 8 or 16 bytes with addresses aligned to their respective lengths. The first consequence is that the length of the `values` and `deltas` arrays must be a multiple of 16 bytes. We solve this by padding these arrays with zeros at the end, which for non trivially large matrices adds only a negligible constant overhead. Because MACKO does not put any constraints on the number of non zeros in each row, the first `load_size` elements of a row (with their corresponding deltas) may not be properly memory aligned for vectorized loading. The straightforward solution would be to pad the end of every row. We employ a more efficient technique of reverse offset memory alignment (ROMA)(Gale et al., 2020). After loading the row offset and calculating the row

length, each thread decrements its row offset to the nearest vector-width-aligned address and updates the number of nonzeros that it needs to process. To maintain correctness, threads mask any values and deltas that were loaded from the previous row prior to accumulating the result in the first iteration of the main loop. Other GPU optimizations are mentioned in Section D.

---

**Algorithm 1** Exclusive prefix sum using warp shuffling

---
**Input:** current thread `lane` holds `local_sum`
Initialize `prefix_sum = local_sum`.
**for** (i = 1; i < 32; i *= 2) **do**
  sync = __shfl_up_sync(prefix_sum, i)
  **if** `lane >= i` **then**
    prefix_sum = prefix_sum + sync
  **end if**
**end for**
**Return:** `prefix_sum - local_sum`

---

## 5. Performance Evaluation

We assess the performance of MACKO at two levels: (1) measuring pure SpMV runtime across a variety of matrix sizes, sparsity levels, and GPU models, (2) end-to-end LLM inference using MACKO-SpMV instead of linear layers.

### 5.1. Runtime

Following the methodology of SpInfer-SpMM (Fan et al., 2025), we evaluate MACKO on random matrices using a diverse set of matrix sizes derived from popular LLM models. These include the Llama2 series (Touvron et al., 2023), the Llama3 series (Dubey et al., 2024), the OPT-Series (Zhang et al., 2022), Qwen2 (Team et al., 2024), and the Mixtral-8×7B MoE model (Jiang et al., 2024).

We evaluate MACKO across a range of sparsity levels against (1) dense representation and cuBLAS library provided by GPU vendor (NVIDIA Corporation, 2025a) (2) cuSPARSE, sparse library provided by vendor (NVIDIA Corporation, 2025b) (3) DASP, state-of-the-art SpMV library (Lu & Liu, 2023) (4) Sputnik, state-of-the-art SpMM library (Gale et al., 2020). The evaluation is conducted for the sparsity levels between 0% and 95% for completeness, even though the most interesting range for pruning is between 30% and 90%.

The SpMV experiments are performed on three different NVIDIA GPUs with different Compute Capabilities (CC): RTX 2080 SUPER (CC 7.5, 8 GB memory), RTX 3090 (CC 8.6, 24 GB memory) and RTX 4090 (CC 8.9, 24 GB memory). We measure the multiplication time in $\mu s$, averaged over 1000 runs with 100 iterations for warm starting. To model the LLM inference where different weights need to

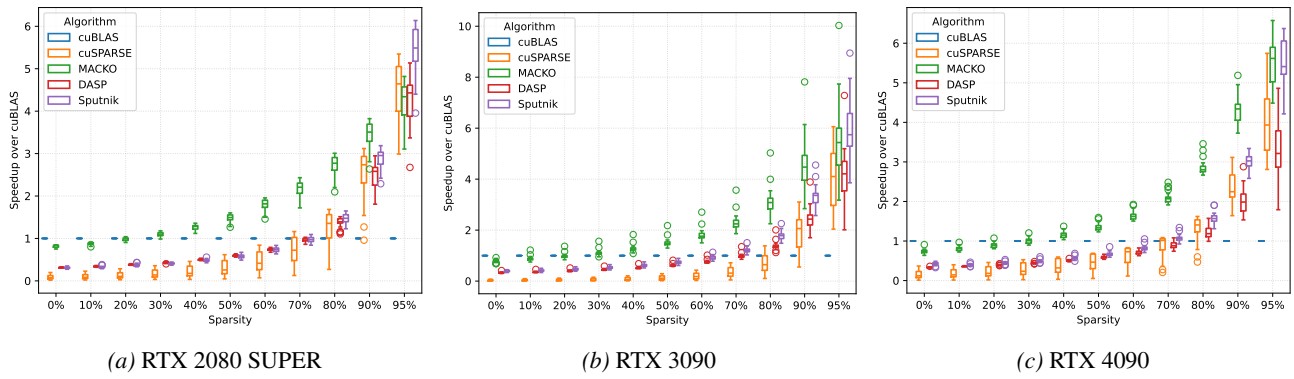

*Figure 4.* Relative speedup over cuBLAS across different matrix sizes.

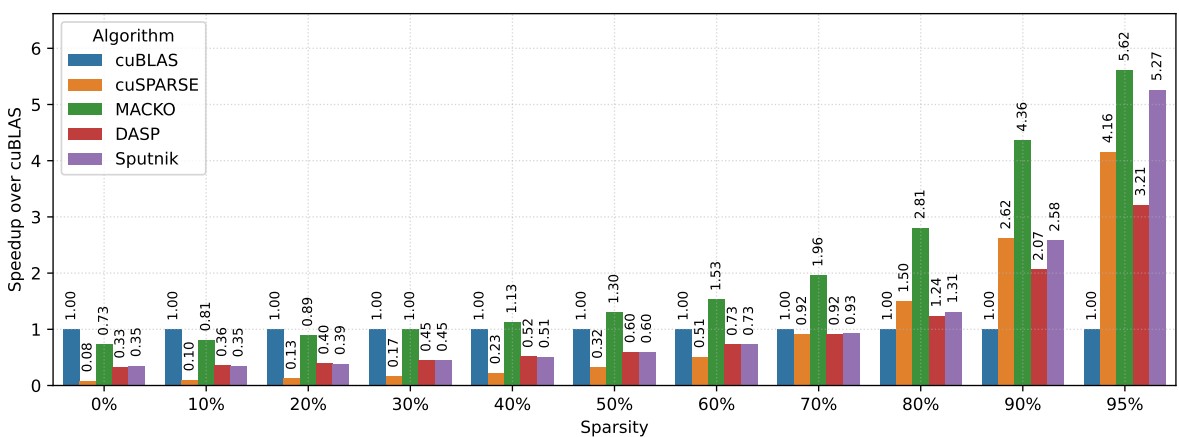

*Figure 5.* Speedup of MACKO relative to cuBLAS on RTX 4090 for $12288 \times 12288$, cuBLAS runtime 336 $\mu s$.

be loaded from GPU memory, we invalidate the GPU cache before every run.

Figure 4 shows performance on different GPU models. Relative speedup values are normalized by cuBLAS runtime. MACKO consistently outperforms cuSPARSE for all GPUs and all measured sparsity levels, except for the extreme sparsity of 95% where it is also outperformed by Sputnik. MACKO matches the performance of dense cuBLAS for sparsity, often as low as 20% and consistently at 30%. MACKO outperforms all baselines across all GPUs for all sparsities between 30% and 90%. This is impressive as MACKO does not introduce any tunable parameters and works out-of-the-box for all matrix sizes, sparsity levels, and GPU models.

Figure 5 shows detailed performance on RTX 4090 for matrix size $12288 \times 12288$. MACKO outperforms cuBLAS for all levels of sparsity above 30%, reaching $1.3\times$ speedup at 50% sparsity and $1.96\times$ at 70% sparsity, all the way to $5.62\times$ at 95% sparsity. At 95% sparsity, MACKO starts to be outperformed by Sputnik.

Using MACKO sparse computation exceeds the perfor-

mance of cuBLAS at sparsity as low as 20%. Existing libraries require $2.6\times$ fewer non-zeros to achieve the same performance.

### 5.2. End-to-end LLM inference

We further perform End-to-End evaluation on Llama2-7b (Touvron et al., 2023) pruned to various sparsity levels with Wanda (Sun et al., 2023) pruning algorithm in the unstructured mode. We measure the time and memory requirements to generate 100 tokens from an empty prompt. This is closely aligned with a real world use case that often decouples the prefill and decode phase (Oh et al., 2024; Patel et al., 2024; Zhong et al., 2024).

We compare the dense representation using cuBLAS multiplication with MACKO. Because MACKO is a loss-less format, for a set seed, these two models produce the exact same output up to a rounding error and minor numerical instabilities caused by non cumulative nature of float arithmetic. All experiments are run on RTX 4090.

Table 1 shows that MACKO consistently provides speedup and memory reduction for all sparsity levels above 30%.

*Table 1.* Memory requirements and generation speed of Llama2-7b pruned to different sparsity levels in dense representation compared to MACKO.

| SPARSITY | DENSE SIZE [GB] | MACKO SIZE [GB] | DENSE SPEED [TOKS/SEC] | SPARSE SPEED [TOKS/SEC] |
|---|---|---|---|---|
| 20% | 13.59 | 13.77 | 66.55 | 66.36 |
| 30% | 13.59 | 12.08 | 66.49 | 73.88 |
| 40% | 13.59 | 10.53 | 66.48 | 85.46 |
| 50% | 13.59 | 8.87 | 66.53 | 98.60 |
| 60% | 13.59 | 7.14 | 66.51 | 119.27 |
| 70% | 13.59 | 5.61 | 66.50 | 150.78 |
| 80% | 13.59 | 4.06 | 66.54 | 193.79 |
| 90% | 13.59 | 2.67 | 66.48 | 255.01 |

*Table 2.* Effective density of MACKO on weights produced by different pruning techniques with their. Perplexity does not depend on MACKO, only on the pruning algorithm.

| METHOD | *effd* | PERPLEXITY |
|---|---|---|
| WANDA | 0.6254 | 8.62 |
| ADMM | 0.6256 | 8.02 |
| FISHER | 0.6259 | 8.42 |

For sparsity $50\%$ we observe $1.53\times$ memory reduction and $1.4\times$ speedup. The size of the model is higher than one would expect based on the model sparsity at high sparsity levels because Wanda does not prune the input embedding layer nor the output head. For Llama2-7b, these layers have $262.144$ MB, which is only $2\%$ of the original model, but $10\%$ of the model size pruned to $90\%$ sparsity.

### 5.2.1. EFFECTIVE DENSITY FOR OTHER PRUNING ALGORITHMS

We evaluate the effective density of the MACKO storage format when applied to weights generated by alternative pruning methods, namely ADMM (Boža, 2024) and Fisher pruning (Liu et al., 2021), where weight importance is estimated as the product of the weight magnitude and the diagonal Fisher information. In contrast to Wanda, both ADMM and Fisher pruning enforce sparsity at the layer level rather than per output channel. We conduct the evaluation on Llama3-8B at $50\%$ sparsity. As shown in Table 2, the empirical effective density closely matches the predicted expectation of $0.625$, reaching values only up to $0.6259$ across all pruning methods. We also note that any uniformly distributed pruning pattern at $50\%$ sparsity yields an effective density upper bounded by $0.664$.

### 5.2.2. COMPARISON WITH 2:4 SPARSITY

Unstructured sparsity enables models with low density while maintaining strong model quality. However, evaluations of-

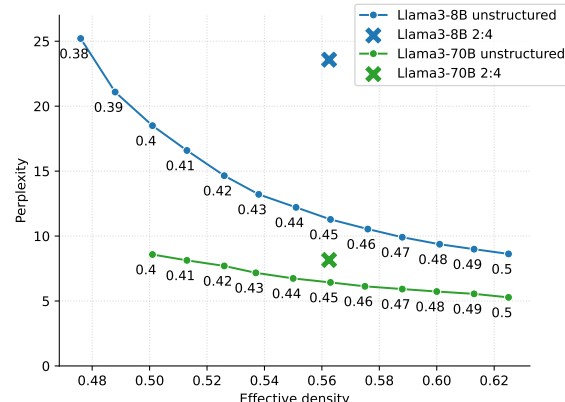

*Figure 6.* The trade-off between perplexity and effective density for unstructured sparsity and 2:4 structured sparsity with Wanda pruning. Each point is annotated with the corresponding model density (contrasted with effective density).

ten omit reporting effective density or speedup. In contrast, structured sparsity is used to obtain predictable speedups and memory reduction, but it is limited to $50\%$ density and leads to significant degradation in model quality.

We compared these pruning approaches on Llama3-8B/70B models (Dubey et al., 2024) using Wanda (Sun et al., 2023) calibrated with 256 RedPajama (Weber et al., 2024) samples. We then evaluated pruned models using WikiText2 (Merity et al., 2016) perplexity.

Fig. 6 demonstrates that unstructured pruning with MACKO format outperforms structured pruning in the perplexity - effective density trade-off. At the same perplexity, unstructured sparsity with MACKO delivers higher memory reduction (lower effective density) and higher $1.8\times$ speedup compared to $1.6\times$. At the same effective density, unstructured sparsity achieves significantly better perplexity: $6.43$ compared to $8.15$ for 70B model and $11.28$ compared to $23.58$ for 8B model. [5]

## 6. Extensions and Current Limitations

Theoretically, MACKO can achieve $1.6\times$ speedup at sparsity $50\%$, while our current implementation achieves $1.2 - 1.5\times$ speedup. The biggest possible improvement of MACKO is to introduce hardware specific optimization. Under optimal setup it is possible to achieve the same memory bandwidth as cuBLAS which will translate to $\frac{1}{effd}$ speedup.

The second possible venue for improvement is optimization for higher sparsity levels above $95\%$, where MACKO is outperformed by Sputnik. At these levels, the value padding becomes the dominant driver of effective density and memory bandwidth is further used by sparse access to values

---

[5]See Section F for details on 2:4 sparsity speedups.

from vector $V$.

Thirdly, we hope to extend this work in the future to accommodate the combination of pruning and quantization. While MACKO shows a promising effective density even for low precisions of $b_{val} = 8$ and a viable effective density for $b_{val} = 4$, more optimization is needed to support efficient SpMV in these settings.

## 7. Conclusion

In this paper, we introduced MACKO, a novel format for representing sparse matrices at low sparsity levels accompanied by an efficient SpMV implementation for GPUs using this format. We identified effective density as a primary bottleneck in SpMV and proposed a solution based on coordinate compression and strategic padding. The experimental results show that MACKO brought significant memory reduction and speedup over all SpMV baselines as well as a dense alternative for sparsity between $30\%$ and $90\%$. As seen in Figure 1, MACKO breaks the barrier of practical usability for sparsity $50\%$. Future work will aim to extend these techniques to lower precisions (8-bit and 4-bit values), matrix-matrix multiplication with small batch dimensions and across even wider range of GPU models.

## Software and Data

The MACKO-SpMV library is open-source and available at github.com/vlejd/macko_spmv. It includes implementation of the MACKO-SpMV cuda kernel, matrix conversion utilities, as well as integration into the `torch` library with enabled torch compilation.

## Acknowledgements

The authors thank Vast.ai for providing easy, convenient, and reliable access to a wide variety of different GPUs. The authors thank Zhen Ning David Liu, Filip Hanzely, Marek Šuppa, and Barbora Klembarová for constructive criticism of the manuscript. The authors thank Miroslav Psota for help with kernel compilation setup.

This research was supported by funding from the Slovak Research and Development Agency grant APVV-24-0045 and by grants 1/0140/25, and 1/0538/22 from the Slovak research grant agency VEGA. Part of the research results was obtained using the computational resources procured in the national project National competence centre for high performance computing (project code: 311070AKF2) funded by European Regional Development Fund, EU Structural Funds Informatization of society, Operational Program Integrated Infrastructure. We acknowledge the EuroHPC Joint Undertaking for awarding this project access to the EuroHPC supercomputer LEONARDO, hosted by CINECA (Italy) and the LEONARDO consortium through an EuroHPC Benchmark and AI Access calls.

## Impact Statement

This paper presents work whose goal is to advance the field of Machine Learning. This work introduces a sparse matrix format and GPU kernel that significantly improve the efficiency of sparse matrix multiplication for pruned neural networks. By enabling high performance on commodity GPUs, our approach lowers the hardware barrier for deploying sparse models, making pruning a more practical and accessible optimization technique. This can reduce computational cost and energy consumption while broadening access to efficient deep learning on consumer hardware.

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

## A. Notation

- input vector: $V$

- input matrix: $M$

- output vector: $Y = MV$

- number of non zeros in $M$: $nnz$

- number of rows and columns in $M$: $R, C$

- density of $M$: $d = \frac{nnz}{R \cdot C}$

- sparsity of $M$: $s = 1 - d$

- effective density $effd$ of a storage format: $effd = \frac{real\_storage}{R \cdot C \cdot b_{val}}$

- number of bits to store one value in $M$: $b_{val}$

- number of bits to store one delta in $M_{MACKO}$: $b_{\Delta}$

- max delta: $2^{b_{\Delta}}$

- compute intensity: $CI$

- operations per byte ratio: $OPB$

- matrix-vector multiplication: MV

- sparse matrix-vector multiplication: SpMV

## B. LLM Architecture and Inference Process

LLMs are built on transformer architecture (Vaswani et al., 2017; Zhao et al., 2023), which relies on self attention mechanisms and fully connected layers. Input token embeddings are transformed into Query (Q), Key (K), and Value (V) matrices through linear projections. The attention process involves multiplying the Q and K matrices, producing attention scores that weigh the V matrix. Additionally, each transformer layer includes a Feed Forward Network (FFN) that refines token embeddings through two linear transformations with a non-linear activation in between. In case of a pruned LLM model, the Q, K and V projections from the attention part, as well as linear transformations in FFN, become sparse matrix multiplications.

LLM inference consists of two phases: the prefill and decode phases. During the prefill phase, the entire input prompt is processed in parallel via sequence of matrix-matrix multiplications, while the autoregressive decode phase generates tokens sequentially one token at a time via matrix-vector multiplications. The performance of LLMs is limited mainly by the decoding stage due to its sequential nature (Xia et al., 2023).

## C. Limits on optimal storage format for sparse matrices

Every lossless sparse matrix storage format needs to store the element values and enough additional information to reconstruct the sparsity pattern. For matrix of size $n = RC$ and density $d$, there are $\binom{n}{dn}$ different possible sparsity patterns. Assuming that each has the same probability, on average they require at least $log_2(\binom{n}{dn})$ bits to be represented unambiguously. Using identity $lim_{n \to \infty} \frac{log(n!)}{n \, log(n)} = 1$, the amount of bits necessary to store the sparsity pattern with density $d$ is $n(-d \, log_2(d) - (1-d)log_2(1-d))$.

The $effd$ of an optimal sparse storage format becomes

$$effd_{opt} = d + \frac{-d \, log_2(d) - (1-d)log_2(1-d)}{b_{val}}$$

For $b_{val} = 16$, the optimal $effd$ is 0.5625 at density 0.5, 0.3007 at density 0.25 and 0.15897 at density 0.125.

Bitmap encoding achieves this optimal *effd* for density 0.5.

The expected effective density of MACKO with different $b_\Delta$ parameters approaches this optimal *effd* for the respective density levels.

- MACKO with $b_\Delta = 2$ at density 0.5 achieves 0.6
- MACKO with $b_\Delta = 4$ at density 0.25 achieves 0.3157
- MACKO with $b_\Delta = 8$ at density 0.125 achieves 0.1875

## D. GPU optimizations

We make use of various well known cuda optimizations like for loop unrolling, use `const` and `__restrict__` keywords, replacing multiplications with bit shifts where possible and removing most modulo operations. A lot of other optimizations are possible, but become GPU specific. Specifically, the use of asynchronous memory loading, cache bypassing, explicit caching of $V$ and thread block size optimization. These optimizations often introduce new parameters that need to be specifically tuned. For the sake of this paper, we wanted to keep MACKO as general and parameter free as possible.

## E. MACKO with different $b_\Delta$

An interesting mode of the MACKO format is an extreme case where $b_\Delta = 1$. Although in the expected case this format improves memory consumption over dense representation for sparsity as low as 7% the practical improvement seems to have limited applicability.

$b_\Delta = 2$ is ideal for the sparsity between 15% and 45% in terms of storage. However, more research is needed for optimization of the SpMV kernel in this mode.

$b_\Delta = 8$ is a good alternative for very high sparsity above 95%. However, in this sparsity, the main bottleneck of SpMV becomes the scattered access to $V$ and limited number of elements per row that may not saturate the whole warp.

### E.1. Practical implementation of $b_\Delta = 2$

In case of $b_{val} = 16$ and $b_\Delta = 2$, we load 128 consecutive bytes worth of `deltas`, which is sixteen deltas per thread. The problem is that the first 16 threads hold the deltas necessary to perform the first step, and the last 16 threads hold the deltas necessary to perform the second step. We use them across two computational steps and use warp shuffling to correctly redistribute the values to the corresponding threads.

In the first step, the thread $i$ reads the necessary deltas from thread $i/2$ and in the second step from $16 + i/2$. This value shuffling is supported on modern GPUs by `__shfl_sync` for Nvidia GPUs. It requires only the use of registers without the need to access shared or global memory and can be performed as a single instruction.

## F. Benchmarking 2:4 sparsity

Natively 2:4 sparsity requires tensorcore support and is limited to batched matrix multiplication with batch size at least 8. It is advertised as achieving $2\times$ speedup over dense representation, but that is only the case in compute bound regimes. In memory bound regime it achieves $1.6\times$ speedup over dense representation. We measured the speedup in SpMV setting (batch size 1) and observed a dependency on matrix size, with only large matrices (above 20k) achieving the $1.6\times$ speedup. Small matrices often resulted in a slowdown compared to dense representation. We decided to use conservative value of 1.6 (heavily favoring 2:4 sparsity) for our comparison.

## G. List of matrix shapes used for benchmarking

```
[[4096,4096], [8192,8192], [8192,29568], [32000,5120], [32000,8192],
[28672,8192], [5120,5120], [5120,13824], [3584,20480], [4096,11008],
[13824,5120], [18944,3584], [14336,4096], [4096,14336], [8192,28672],
[11008,4096], [32000,4096], [20480,3584], [3584,18944], [21504,7168],
[7168,7168], [28672,7168], [7168,28672], [27648,9216], [9216,9216], [36864,9216],
[9216,36864], [36864,12288], [12288,12288], [49152,12288], [12288,49152]]
```

# H. SpMV runtime

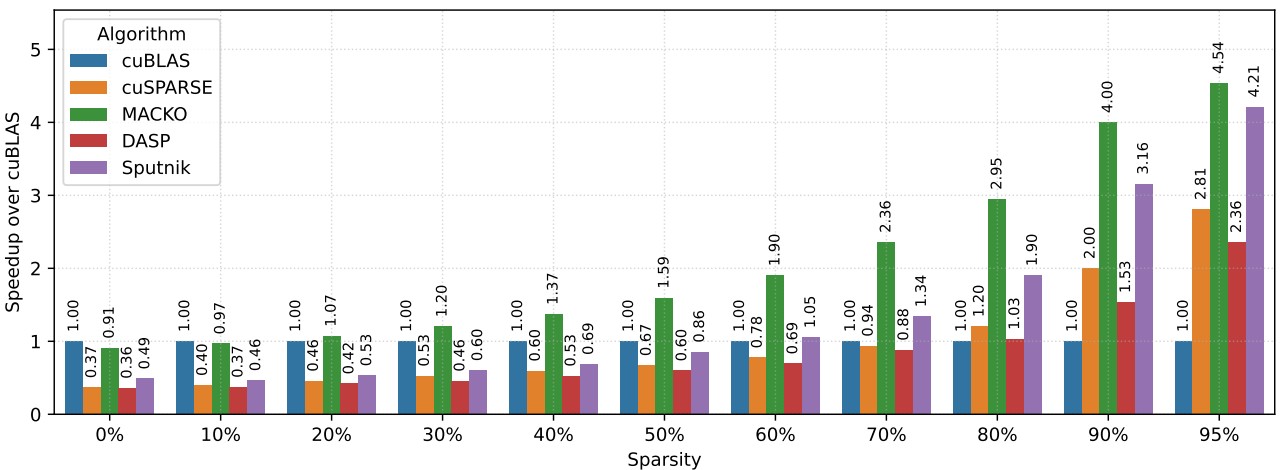

*Figure 7.* Speedup of MACKO relative to cuBLAS on NVIDIA GeForce RTX 4090 for $4096 \times 4096$, cuBLAS runtime $59 \ \mu s$.

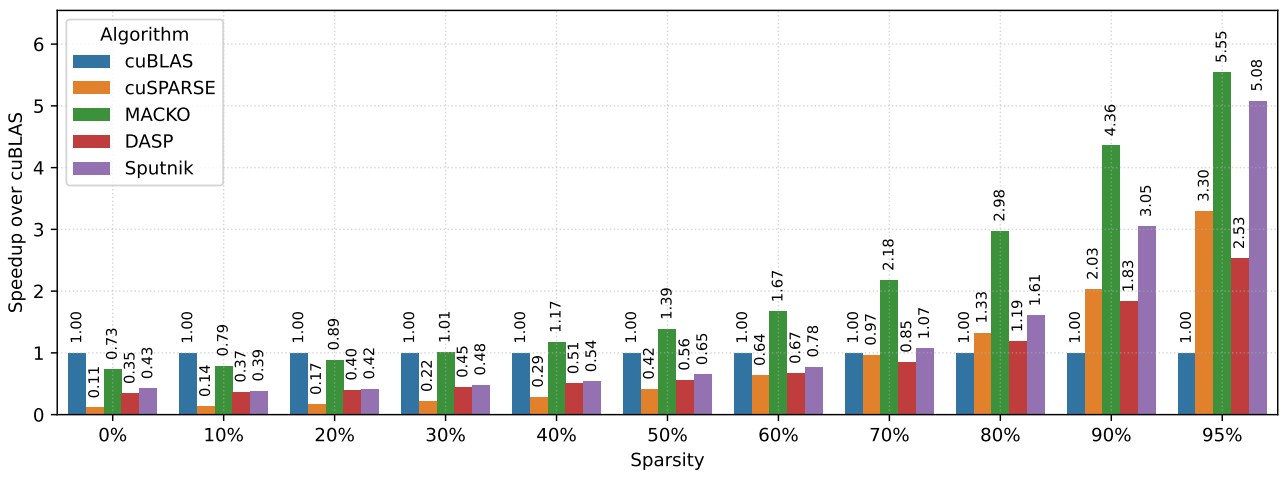

*Figure 8.* Speedup of MACKO relative to cuBLAS on NVIDIA GeForce RTX 4090 for $4096 \times 11008$, cuBLAS runtime $122 \ \mu s$.

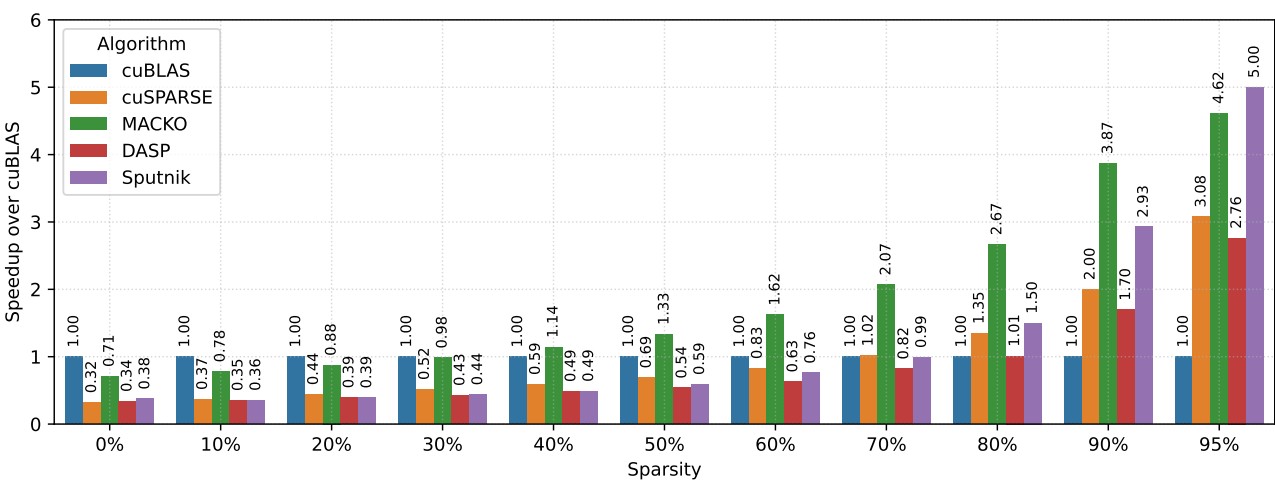

*Figure 9.* Speedup of MACKO relative to cuBLAS on NVIDIA GeForce RTX 4090 for $11008 \times 4096$, cuBLAS runtime 120 $\mu s$.

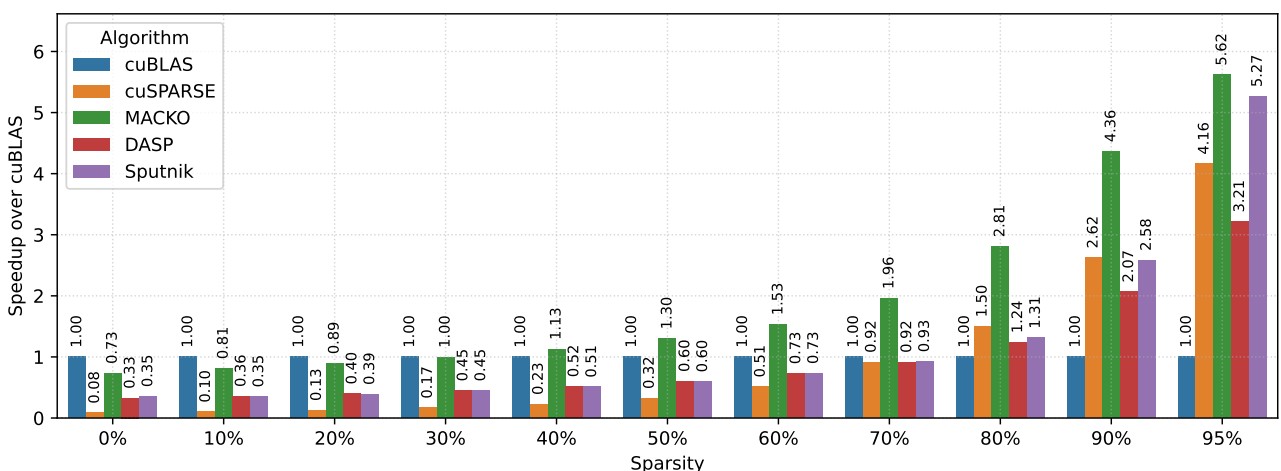

*Figure 10.* Speedup of MACKO relative to cuBLAS on NVIDIA GeForce RTX 4090 for $12288 \times 12288$, cuBLAS runtime 336 $\mu s$.

