# OpenReview forum: "MACKO: Sparse matrix-vector multiplication for low sparsity"
_ICML.cc/2026/Conference — ICML 2026 spotlight_

### Official Review · Reviewer_xP75 · 2026-03-11

**Soundness:** 3
**Presentation:** 3
**Significance:** 3
**Originality:** 3
**Overall Recommendation:** 5
**Confidence:** 4

**Summary:**

This paper addresses the issue of low efficiency of sparse matrix-vector multiplication (SpMV) for large language models (LLM) after pruning under low sparsity (30%-90%). It proposes the DELTA4 format. This method compresses column indices through mutually aligned Delta encoding and introduces a zero-padding strategy to achieve alignment between coordinates and values. It reduces memory bandwidth usage while being compatible with the GPU SIMT architecture. Experiments show that at a 50% sparsity level, DELTA4 achieves 1.5 times memory reduction and 1.2-1.5 times acceleration compared to the dense baseline. It outperforms existing solutions such as cuSPARSE, demonstrating that non-structured pruning is practical on consumer-grade GPUs.

**Compliance With Llm Reviewing Policy:**

Affirmed.

**Final Justification:**

My final recommendation is a Weak Accept (Score: 4). The authors present an ingenious SpMV co-design (DELTA4) that achieves commendable FP16 performance on consumer-grade GPUs. The rebuttal successfully strengthened the paper's soundness by providing the requested empirical evaluations on the Llama3 models and alternative pruning algorithms.

However, my evaluation is tempered by two practical limitations confirmed during the rebuttal. First, the method's performance degrades on high-bandwidth data-center GPUs (H100), indicating that the current design is closely tied to the specific compute-to-memory-bandwidth ratio of consumer hardware. Second, the format requires >50% sparsity to yield memory benefits under 4-bit precision, which limits its immediate compatibility with the rapid industry shift toward INT4/INT8 for efficient local LLM inference.

Despite these constraints, the specific engineering contribution for local FP16 inference is solid and valuable to the community. Therefore, I am leaning towards a Weak Accept. I remain entirely open to raising my score further if the authors can convincingly address these remaining concerns regarding data-center scalability and low-bit quantization in subsequent discussions or the camera-ready revision.

After the author's final comment and answer to questions, I choose to accept (Score: 5).

**Key Questions For Authors:**

1.Since the experiments are conducted primarily on consumer GPUs such as the RTX 4090, it is unclear whether the performance gains reported for DELTA4 in Figure 1 would remain in a similar range on data-center-class GPUs such as the H100, which provide substantially higher memory bandwidth and a different SM architecture.
2.For the end-to-end evaluation, the study only considers the relatively small Llama2-7B model and is restricted to Wanda pruning. It would be important to understand whether the same conclusions hold for larger models, such as Llama3-70B, or for sparsity patterns produced by other widely used pruning methods such as SparseGPT. In particular, under these settings, could the worst-case zero padding substantially slow down decoding?
3.The paper states that the current implementation is mainly optimized for 16-bit values. In an INT8 quantization setting, where the value bit width is reduced by half, the relative cost of incremental encoding and padding would become much larger. Under such a regime, would the alignment overhead offset the bandwidth benefit of the proposed format?
4.Section 6 notes that at very high sparsity levels (>95%), DELTA4 is outperformed by Sputnik because value padding becomes the main factor increasing effective density. Is there any theoretical analysis that characterizes this lower bound? More broadly, does DELTA4 have a fundamental limitation under extremely high sparsity or under certain forms of structured sparsity?

**Limitations:**

yes

**Strengths And Weaknesses:**

Completeness: Advantages: The theoretical derivation is solid, the analysis of effective density is clear, and it proves the boundedness of the fill-in cost in the worst case.
The disadvantage is that it only conducts end-to-end generation tests on the Llama2-7B model and tests only one pruning algorithm, Wanda. The local dense phenomenon in the actual pruning of large models can extremely likely lead to a sharp increase in the fill-in cost in the worst case. The experimental part only used random matrices and the static weights after Wanda pruning, and did not discuss dynamic sparsity or the compatibility of gradient updates during training, which limits its applicability in more extensive scenarios.
Performance: Advantages: The structure is logically sound, the charts are clear, and they visually present the core advantages. The definitions of terms are clear, and the explanation of the format encoding mechanism is also relatively intuitive.
Disadvantages: The drawback lies in the absence of some details of the hardware-level optimizations.
Importance: Advantages: Achieved a breakthrough of 50% unstructured sparsity in FP16 precision, addressing the pain points of unstructured pruning and having certain engineering value.
Disadvantages: Currently, local LLM inference under constrained hardware is highly dependent on 8-bit or lower precision quantization, and DELTA4 clearly indicates that there is currently a lack of optimization support for low precision.
Originality: The idea of combining Delta encoding with the GPU warp-level parallel mechanism is ingenious. Although it is not based on a completely new mathematical principle, by combining it with zero padding to ensure data alignment within GPU thread bundles and unbranched execution, the system-level co-design is highly innovative, distinct from simple format compression.

---

> ### Author Rebuttal · Authors · 2026-03-30
>
> Thank you for recognizing our work as ingenious and highly innovative.
> Here are the responses to your concerns:
> > testing only one pruning algorithm, Wanda and the local dense phenomenon
>
> We have not seen a strong local dense phenomenon. We saw a very small effective density increase (by 1%) for the first attention output projection.
>
> Furthermore, we extended the evaluation to other pruning algorithms, mainly ADMM (https://openreview.net/forum?id=1hcpXd9Jir) and Fisher pruning (where we estimate weight importance using the product of weight magnitude and the diagonal Fisher), and analyzed the effective density of pruned layers.
> Compared to Wanda, ADMM and Fisher pruning do not enforce sparsity per output, but per whole layer. We evaluate it on Llama3-8B using 50% sparsity. The results are very close to the predicted expected case with an effective density 62.59% compared to the expected case with 62.5%. Note that every uniform pruning will have the effective density at most 66.4%.
>
> | Method                        | Effective Density | Perplexity | Per output sparsity |
> | ----------------------------- | ----------------- | ---------- | ------------------- |
> | Wanda (50% sparsity)          | 62.54             | 8.62       | Yes                 |
> | ADMM (50% sparsity)           | 62.56             | 8.02       | No                  |
> | Fisher pruning (50% sparsity) | 62.59             | 8.42       | No                  |
>
> > dynamic sparsity, gradient update
>
> We have not seen DST applied to LLM training yet, but many methods, which alter mask during training, are compatible with Delta4. For example, for Rigging the lottery (https://arxiv.org/abs/1911.11134), one would need to recompute Delta4 encoding after changing the mask, which does not happen that often. In the case of DST or Sparse Momentum, using any sparse format is not easily applicable since both methods keep the dense weights under the hood.
> Regarding the gradient update, Delta4 is directly compatible with weight updates and with infrequent sparsity structure updates. However, training is usually compute bound and relies on matrix-matrix multiplication instead of a matrix vector multiplication which makes Delta4 less beneficial. These issues can be addressed in a same way as with the prefill phase, see our other comments on prefill phase.
>
> > details of the hardware-level optimizations
>
> We will provide more details in the Appendix of the next version of the paper.
>
> > 8-bit or lower precision quantization
>
> Ideas from Delta4 are directly extensible to naive 8bit values. The effective density of 50% sparsity will be 75% (1.3x speedup) at the best case, and 80% at the worst case. For 4 bit values the sparsity would need to go beyond 50% to see any memory reduction.
> We measured Delta4 with 8 bit values to achieve 1.15x - 1.33x speedup and will extend the paper with these results..
> For LLM inference, lower bit formats often introduce additional scaling parameters per whole tensor, output or smaller groups. Besides implementational complexity, the overhead of these parameters stays the same regardless of sparsity and further increases the effective density. These issues may be addressable with further research.
>
> > performance on data-center-class GPUs such as the H100
>
> We evaluated DELTA4 on the H100: memory reduction is identical, and speedup remains competitive (1.0–1.3× at 50% sparsity vs. 1.3–1.5× on consumer GPUs), with a break-even point between 30-50% sparsity (vs. 25% on the 4090). The gap reflects that data-center GPUs have higher memory bandwidth, raising the bar for memory-bound kernels. Our primary target is local consumer-GPU inference; data-center optimization is left for future work.
>
> > conclusions hold for larger models, such as Llama3-70B, or for sparsity patterns produced by other widely used pruning methods such as SparseGPT
>
> We report effective densities for Llama3-70B in Figure 6. As for sparsity patterns produced by other algorithms, we reported them in the answer above about local dense phenomenon.
>
> > performance at very high sparsity levels (>95%)
>
> At high sparsity levels, Delta4 efficiency decreases (due to a high need for padding). In these cases, having 8 bits per delta would be more beneficial. But most LLM pruning algorithms target low-to-mid sparsity, where having 4 bits per delta is better.
>
> Thank you again for your very good suggestions. We hope our response clarifies your concerns. If that’s the case, we would greatly appreciate it if you would consider raising your score.

---

> > ### Author Rebuttal · Reviewer_xP75 · 2026-04-01
> >
> > Thank you for the detailed rebuttal and the additional experimental results. The system-level co-design and the reported gains on consumer GPUs such as the RTX 4090 are appreciated. However, the response raises an important concern about the generality and practical completeness of the proposed method.
> >
> > In particular, the authors acknowledge that the speedup degrades on data-center GPUs such as the H100, while optimization for such platforms is deferred to future work. This is concerning because large-scale LLM inference is predominantly deployed on high-bandwidth data-center hardware. As a result, the current evidence suggests that DELTA4 may be closely tied to the compute-to-memory-bandwidth characteristics of consumer GPUs, rather than providing a broadly applicable sparse format.
> >
> > In addition, the discussion on low-bit quantization further limits the practical scope of the method. The rebuttal indicates that 4-bit precision delivers memory benefits only when sparsity exceeds 50%, and the reported speedup for 8-bit values remains relatively marginal. This weakens the connection of DELTA4 to current deployment trends, where INT4/INT8 quantization is increasingly crucial for efficient local and edge LLM inference.
> >
> > Overall, these points make it difficult to assess the broader practical significance and architectural completeness of DELTA4 in its current form. I remain open to raising my score if the authors can more convincingly address these concerns about applicability and maturity.
> >
> > Q1: How can DELTA4 substantiate its broad practical significance and architectural completeness if its benefits diminish on data-center GPUs such as the H100, and its memory and speed advantages under low-bit quantization remain heavily constrained?

---

> > > ### Author Response · Authors · 2026-04-06
> > >
> > > > H100
> > >
> > > We appreciate the reviewer raising this point, as it gives us the opportunity to clarify an important distinction between algorithmic generality and implementation completeness on a specific hardware target.
> > >
> > > We want to make it clear that the algorithm's performance on H100 is not diminished, the current implementation does not yet extract the theoretical maximum on H100. Even in the current unoptimized version, we consider 1x-1.3x speedup and 1.5x memory reduction at 50% unstructured sparsity to be a significant improvement over dense representation (which is the strongest baseline) diminished only by the fact that we were able to achieve even better results on consumer GPUs.
> > >
> > > Matrix vector multiplication is still memory bound on H100 (2TB/s, 200 fp16 FLOPS -> 100ops/byte) and all our claims about benefits of DELTA4 hold just the same, if not more. However the H100 architecture (Hopper) introduced a fundamentally different kernel programming model. To obtain optimal memory throughput, warps must be explicitly partitioned into producer and consumer roles with synchronization through shared-memory barriers and utilization of asynchronous memory loading operations.
> > > Porting a kernel to Hopper will not change the algorithm or the underlying ideas behind DELTA4, it is a structural reimplementation.
> > >
> > > We would like to draw a parallel with FlashAttention:
> > > - FA v1 demonstrated strong results on a single GPU architecture without mentioning any results on Hopper (even though the existence of the architecture is specifically cited).
> > > - FA v2 acknowledged that FA provides benefits on H100, but required a "different implementation" that is estimated to achieve 2x speedup and deferred it to later work
> > > - FA v3 eventually delivered the Hopper-native implementation as a standalone paper.
> > > - FA v4 recently delivered implementation for Blackwell-based systems such as the B200 and GB200
> > >
> > > We are in the same position as FA at the v1/v2 stage. The algorithm's theoretical gains are if anything larger on H100 due to its higher memory bandwidth.
> > > The engineering challenge is that realizing them requires a Hopper-native implementation, which we explicitly defer to future work. We believe this is the appropriate and honest scoping of a single paper.
> > >
> > > > Low bit quantization interaction
> > >
> > > We completely agree with the reviewer's concerns for the low-bit regime. As we show in the Appendix C, this is an inherent limitation of unstructured sparsity in general, not of our algorithm or storage format. For 4bit values, and 50% sparsity, DELTA4 extension with 2bit deltas achieves expected effective density 0.8 (1.25x memory reduction), which is very close to the information theory bound on optimal effective density 0.75 (1.33x memory reduction).
> > > DELTA4 brings this limitation to light and can achieves a memory reduction and a speedup that is close to this theoretical limit.

---

### Official Review · Reviewer_hZnG · 2026-03-11

**Soundness:** 3
**Presentation:** 3
**Significance:** 3
**Originality:** 2
**Overall Recommendation:** 4
**Confidence:** 3

**Summary:**

The paper proposed DELTA4, a sparse matrix format and GPU kernel for SpMV in the low-sparsity regime for pruned LLM inference. The primary idea is to compress column coordinates using delta encoding, combined with padding and alignment. The results suggest meaningful speedups over prior sparse baselines in certain low-sparsity settings, and improvements over dense FP16 inference in the presented decode workload.

**Compliance With Llm Reviewing Policy:**

Affirmed.

**Final Justification:**

The added profiling and empirical effective-density results improve my confidence in the paper. In particular, the kernel profiling makes the proposed mechanism more convincing, and the new real-layer measurements address my concern that realistic pruning patterns might cause substantially worse padding overhead than the expected-case analysis suggests.

The rebuttal also clarifies the novelty better: I now view the contribution more as a GPU-oriented co-design of fixed-width delta encoding, padding, alignment, and warp-friendly execution, rather than delta coding alone.

I still see some scope limitations. The strongest evidence remains in the decode-phase batch-1 setting, and while the rebuttal gives a convincing explanation of the prefill story, the practical speedup there still appears limited, especially when prefill is dominant. Applicability to lower-bit settings also remains limited at present.

Overall, however, the rebuttal resolves my main technical concerns sufficiently for me to raise my score to Weak Accept.

**Key Questions For Authors:**

1. Can you  provide kernel-level profiling to validate the proposed mechanism behind DELTA4’s speedups? A time breakdown across delta decoding, coordinate reconstruction, dense-vector gathers, and reductions would help.

2. Can you report empirical padding/effective-density statistics on real pruned LLM layers?

3. Can you comment on whether DELTA4 is compatible with speculative decoding?

**Limitations:**

yes

**Strengths And Weaknesses:**

**Strengths:**

1. The paper targets an important and practically relevant regime: low-to-moderate sparsity, where many existing sparse formats and kernels provide limited benefit in sparse LLM inference.

2. The empirical results suggest that the proposed method is competitive and often clearly stronger than prior sparse SpMV baselines in the regime the paper targets.

3. The paper is generally well motivated and the high-level intuition is easy to follow, reducing coordinate overhead can matter substantially in the low-sparsity regime, where index storage can dominate the gains from sparsity.

---

**Weaknesses:**


1. The paper uses delta encoding as one of the  main ingredients, but does not sufficiently position this choice relative to prior delta-coded sparse representations [R1]. Since delta coding of sparse indices is a known idea, the paper should cite the relevant prior formats and explain more explicitly what is new here.

2. The evaluation emphasizes decoding, but practical LLM inference also depends on prefill latency. The paper doesn’t show whether the reported gains translate to end-to-end performance when prefill is included.

3. Since Table 1 is intended to support practical LLM inference relevance, it would be helpful to compare against at least one competitive sparse inference baseline, or clarify why such a comparison is infeasible.

4. The paper would benefit from a more complete practical tradeoff analysis. Since the goal is deployable sparse LLM inference, reporting model quality together with memory and decoding speed, would provide a clearer picture of the real deployment.

5. The effective-density (padding) analysis assumes i.i.d. uniformly distributed nonzeros. Since DELTA4 depends on bounded deltas and greedy zero-padding, realistic pruned LLM sparsity could significantly increase padding, potentially nearing the worst case. The paper does not provide sufficient evidence to show the claimed benefits reliably generalized.

[R1] Trommer, Elias, Bernd Waschneck, and Akash Kumar. "dCSR: A memory-efficient sparse matrix representation for parallel neural network inference." 2021 IEEE/ACM International Conference On Computer Aided Design (ICCAD). IEEE, 2021.

---

> ### Author Rebuttal · Authors · 2026-03-30
>
> We thank the reviewer for the detailed feedback and address each point below.
> > prior delta-coded sparse representations
>
> Delta coding is well-known, but has only been successfully efficiently implemented on CPUs [R1]. [R1]'s format is more complex, encoding differences from predicted, rather than previous, positions.
> Works that target the CPU generally rely on a computation that is not compatible with the level of parallelism that GPUs provide (the need to parallelize across both dimensions of the matrix) or with their constraints (SIMD warp model and collaborative memory access). The strongest CPU baseline, delta entropy coding, relies on dynamic bitwidth of encoded deltas, which makes the amount of work within a warp imbalanced and inefficient. The second strong baseline, bitmasking the nonzero elements, introduces conditional memory loading that can not be efficiently done on a GPU.
>
>
> > prefill latency
>
> DELTA4 targets the memory-bound decode phase (batch-1 matrix-vector multiplication) and does not accelerate prefill.
>
> In general, there are 2 approaches to addressing the prefill. “Decoupling” keeps a copy of the weights in a more matrix-matrix multiplication-friendly format (dense representation). Assuming large context size, the prefill is compute-bound even if we load the model weights from host memory. This increases the requirements on the host memory but keeps the prefill latency the same as in the dense setting.
> “Load as Sparse, Compute as Dense”, coined by FlashLLM, dynamically decodes the sparse matrix into dense representation. We argue that DELTA4 can be efficiently decoded on the GPU into a dense representation and estimate that with optimal pipelining, there could be around 12% slowdown in the prefill phase (based on SpInfer), but keep the memory improvements.
>
> Assuming the dense baseline spends 5% time on prefill and 95% time on generation, a 50% sparse network would achieve a 1.42x speedup (compared to the reported 1.48x for generation only)
>
> For use cases where prefill is dominant, we can not offer any speedup. We may offer a memory reduction, which may have an impact on a limited number of use cases.
>
>
>
> > compare against at least one competitive sparse inference baseline, or clarify why such a comparison is infeasible
>
> Popular unstructured sparse inference libraries (DeepSparse, llama.cpp) target CPUs and are not meaningful GPU baselines. Flash-LLM requires large batches and Tensor Cores, which do not support matrix-vector multiplication. The most relevant GPU baseline is hardware-accelerated 2:4 structured sparsity, which we compare against in section 5.2.1. Notably, our purely algorithmic approach with no fine-tuning is competitive with a method backed by years of hardware development. If you have a specific unstructured SpMV GPU kernel in mind, we are happy to evaluate it.
>
> If you have a specific unstructured SpMV GPU baseline in mind, we are happy to evaluate it.
>
> > reporting model quality together with memory and decoding speed
>
> Figure 6 shows the quality/size tradeoff. We will add these metrics to Table 1.
>
> > realistic pruned LLM sparsity could significantly increase padding, potentially nearing the worst case
>
> Section 4.2 (L244) derives the effective density upper bound, the primary runtime driver.
> For fp16 values, 4-bit deltas, and 50% sparsity, worst-case effective density is 66.4% vs. 62.5% (best) and 62.501% (expected) - only a 4% worst-case increase.
> Measured worst case adversarial patterns still yield 1.45× speedup (vs. 1.55× expected) and 1.5× memory reduction (vs. 1.6× expected).
> We also extended evaluations to other pruning algorithms (see Reviewer xP75).
>
> > kernel-level profiling
>
> Profiling confirms the kernel is memory-bound: 99.6% of warp stall time is Long Scoreboard (memory latency), with 54% of time loading deltas and 24% loading values. Memory throughput is 81.8% of peak (theoretical max ~95%), leaving limited headroom. Compute throughput is 28.7%, confirming computation is not the bottleneck.
>
> > empirical padding/effective-density statistics
>
> With 50% Wanda sparsity, effective density is 62.55% on most layers, and 62.61% on average for output attention projections (worst case: 63.51% on the first attention output projection).
>
> > Can you comment on whether DELTA4 is compatible with speculative decoding?
>
> Speculative decoding transforms the inference process from matrix vector multiplication to matrix-matrix multiplication.
> DELTA4 currently supports only matrix vector multiplication. Using it for batched inference scales linearly, requiring >65% sparsity (batch 2) or >90% (batch 4) to match dense inference speed, though memory benefits remain.
>
> Extension to small batches without linear scaling is possible, but it poses multiple practical challenges, mainly related to register pressure and low occupancy.
>
> We hope this addresses your concerns. We welcome the opportunity to discuss further and hope you will consider updating your score.

---

> > ### Author Rebuttal · Reviewer_hZnG · 2026-04-02
> >
> > Thank you for the detailed rebuttal. The added profiling and empirical effective-density results improve my confidence in the paper. In particular, the kernel profiling makes the proposed mechanism more convincing, and the new real-layer measurements address my concern that realistic pruning patterns might cause substantially worse padding overhead than the expected-case analysis suggests.
> >
> > The rebuttal also clarifies the novelty better: I now view the contribution more as a GPU-oriented co-design of fixed-width delta encoding, padding, alignment, and warp-friendly execution, rather than delta coding alone.
> >
> > I still see some scope limitations. The strongest evidence remains in the decode-phase batch-1 setting, and while the rebuttal gives a convincing explanation of the prefill story, the practical speedup there still appears limited, especially when prefill is dominant. Applicability to lower-bit settings also remains limited at present.
> >
> > Overall, however, the rebuttal resolves my main technical concerns sufficiently for me to raise my score to Weak Accept. For the revised version, it may also be helpful to briefly discuss whether DELTA4 could have any role in speculative decoding on the drafter side, where computation may still be closer to a batch-1 or low-batch autoregressive regime.

---

> > > ### Author Response · Authors · 2026-04-06
> > >
> > > > role in speculative decoding on the drafter side
> > >
> > > Thank you very much for clarification, we completely missed this application and under appreciated the first suggestion on speculative decoding.
> > >
> > > The existence of DELTA4 significantly moves the Pareto front of speed/accuracy tradeoff for all unstructured LLM pruning approaches.
> > > We need to decompose this into two parts: speed/sparsity and sparsity/accuracy.
> > > The sparsity/accuracy tradeoff depends solely on the pruning algorithm, while DELTA4 unilaterally improves the speed/sparsity.
> > >
> > > Assume we are using Llama-2-70B as the main model and Llama-2-7B as the drafter.
> > > There are two main potential improvements.
> > > - Using a pruned version of the large mode as a drafter. Assuming a drafter pruned to 60% sparsity has acceptance rate 0.9 (optimistic, but not unreasonable as these models have very large overlap), the cost coefficient goes down to 0.5, and we can expect ~1.3 x speedup. This is a significant speedup, as the drafter can be created very easily with post training pruning, and even be conditioned on a specific problem subset.
> > > - Using pruning on existing drafter. Using 0 shot accuracy as a proxy for acceptance rate, pruning a drafter to 50% sparsity would yield acceptance degradation from 59.67% to 57.73% and the cost coefficient decreases from 0.1 to 0.067. This yields approximately 6% speedup for 4 token long proposals.

---

### Official Review · Reviewer_sU3C · 2026-03-16

**Soundness:** 3
**Presentation:** 4
**Significance:** 4
**Originality:** 3
**Overall Recommendation:** 5
**Confidence:** 4

**Summary:**

The authors present DELTA4, a new data storage format to do SpMV in the range of 30-90% sparse matrices. This is specifically relevant for LLMs, as mostly they are pruned till 50% sparsity but in a structured manner, leading to better runtime but poor generalization. With DELTA4, the authors claim that the models can be pruned to 50% unstructured sparsity which can give better generalization performance without sacrificing runtime. The authors provide a custom kernel implementation that is integrated in PyTorch, making it easy to use technique.

**Compliance With Llm Reviewing Policy:**

Affirmed.

**Final Justification:**

I thank the authors for their response. I think Delta4 is a nice technique which can potentially have larger impact than just LLMs. I believe my current score reflects my opinion of the work. Hence, I will keep the current score.

**Key Questions For Authors:**

1. One of the recent works in this domain is SMAT Library (https://arxiv.org/pdf/2408.11551). I am wondering if authors have some comments on the comparisons with this method? I understand that doing more experiments is probably not feasible but I, as a reader would appreciate some comparison in the text.

2. If I understand correctly, the encoding has to take place offline with DELTA4 which limits its applications to say scenarios like Dynamic Sparse Training (DST). While that is not the focus of the paper, I think it will add to the paper if readers can know what are the cost or challenges associated with developing a method like this for DST as that can fundamentally help a orthogonal field of research.

3. The 'expected case' effective density relies on the assumption that non-zeros are uniformly and independently distributed. Did you analyze the actual distribution of zeros produced by the Wanda algorithm? Does Wanda tend to create clustered non-zeros or long contiguous blocks of zeros that push the padding overhead closer to the 'worst case' scenario?"

**Limitations:**

Yes.

**Strengths And Weaknesses:**

**Strengths**
1. The authors show improvements on end-to-end inference tasks on models of decent size, showing the effectiveness of the method. Moroever, the improvements in execution time can be noticed at lower sparsities as compared to other methods that require > 50% sparsity.

2. The approach presented in the paper is quite neat as it does not require any parameter finetuning, offline computation etc and hence can probably work out-of-the-box, making it easy to adapt.

3. The fact that DELTA4 works out-of-the-box across generations of GPUs shows that its core philosophy is fundamentally sound.

**Weaknesses**

1. As authors have mentioned, current approach relies on FP16 format, which is probably not as used as more quantized versions of LLMs, which is a major application of this approach at this point of time. I do however think that it is not a major drawback of the approach but rather something to remember.

2. If LLMs are an application of concern, I would appreciate some comments in the text about if the authors think that their approach can be extended to the prefill phase of the computation as well. If not, what could be the challenges there.

---

> ### Author Rebuttal · Authors · 2026-03-30
>
> Thank you for a helpful review and suggesting important extensions to our work.
> Here are the responses to your concerns:
> > current approach relies on FP16 format
>
> Ideas from Delta4 are directly extensible to naive 8bit values. The effective density of 50% sparsity will be 75% (1.3x speedup) at the best case, and 80% at the worst case. For 4 bit values the sparsity would need to go beyond 50% to see any memory reduction.
> We measured Delta4 with 8 bit values to achieve 1.15x - 1.33x speedup at 50% sparsity and will extend the paper with these results.
> For LLM inference, lower bit formats often introduce additional scaling parameters per whole tensor, output or smaller groups. Besides implementational complexity, the overhead of these parameters stays the same regardless of sparsity and further increases the effective density. These issues may be addressable with further research.
>
> > Can the approach be extended to the prefill phase of the computation, comment on prefill latency
>
> In short, this approach will not speed up the prefill phase and hence will have a very limited impact in use cases where prefill is dominant.
> Our approach was designed to specifically target the memory-bound decoding phase and not the compute-bound prefill phase. This is consistent with the trend of increased reasoning and agentic use cases. For simplicity, we assumed that prefill could be decoupled from the decoding phase (L373, right column).
>
> In general, there are 2 approaches to addressing the prefill: decoupling and “load as sparse, compute as dense”. The “decoupling” approach keeps a copy of the weights in a more matrix-matrix multiplication-friendly format (for example, dense representation). Assuming sufficiently large context size, the prefill is compute-bound even if we load the model weights from host memory. This approach increases the requirements on the host memory (generally considered cheap) but keeps the prefill latency the same as in the dense setting.
> The second approach, “Load as Sparse, Compute as Dense”, coined by FlashLLM and used by SpInfer, is to dynamically decode the sparse matrix, in our case in DELTA4 format, into dense representation. Here, the Delta4 would decrease the GPU memory requirements of the prefill phase. We argue that DELTA4 can be efficiently decoded on the GPU into a dense representation and estimate that with optimal pipelining, there could be around 12% slowdown in the prefill phase (based on SpInfer). The final end-to-end speedup will depend on the context size for the prefill, the number of generated tokens and the compute/memory throughput of a specific GPU.
>
> Assuming the dense network would spend 5% time on prefill and 95% time on generation, a network pruned to 50% would achieve a 1.42x speedup (compared to the reported 1.48x for generation only)
>
> For use cases where prefill is dominant, we can not offer any speedup. We may offer a memory reduction, which may have an impact on a limited number of use cases.
>
> > SMAT Library comparison
>
> SMAT targets block sparsity (with block size 16x8) SpMM, with complicated preprocessing involving row shuffling, and is better at 78% or 96% sparsity depending on the batch size.
> In a separate project, we tested block sparse pruning, and the degradation of quality is very rapid even with blocks of size 1x4 (we already got PPL above 1000). So we do not think that this is a viable way for LLM pruning.
>
> > dynamic sparse training
>
> For many methods, which alter mask during training, the Delta4 is not a limiting factor. For example, for Rigging the lottery (https://arxiv.org/abs/1911.11134), one would need to recompute Delta4 encoding after changing the mask, which does not happen very often. In the case of DST or Sparse Momentum, using any sparse format is not easily applicable since both methods keep the dense weights under the hood.
> >  actual distribution of zeros produced by Wanda
>
> We analyzed Llama3-8B with 50% Wanda pruning. It seems that in all layers except o_proj, the pattern looks random (i.e., delta is one at 50% times as expected). Attention output projection produces slightly clustered patterns, i.e., consecutive nonzero elements (delta=1) appear more more often than predicted 50%, but in all cases except the first output projection, it is still less than 55%, in the case of the first output projection, it is 68%..
>
> Thank you again for your very good suggestions. We hope our response clarifies your concerns. If that’s the case, we would greatly appreciate it if you would consider raising your score.

---

> > ### Author Rebuttal · Reviewer_sU3C · 2026-04-04
> >
> > I thank the authors for their response. I think Delta4 is a nice technique which can potentially have larger impact than just LLMs. I believe my current score reflects my opinion of the work. Hence, I will keep the current score.

---

### Decision · Program_Chairs · 2026-04-30

**Decision:**

Accept (spotlight)

**Comment:**

This paper investigates efficient SpMV kernels for the low-sparsity unstructured pruning regime, which is highly relevant in practice. Existing pruning methods for LLMs typically produce 30%–90% unstructured sparsity, yet under this workload, existing SpMV approaches often perform poorly and fail to deliver sufficient gains in either memory footprint or runtime efficiency.
To address this challenge, the paper proposes DELTA4, a GPU-oriented co-design of sparse data format and kernel implementation. The method is built on three main ideas: delta encoding for compressing column indices, padding and alignment to better match the GPU warp/SIMT execution model, and explicit analysis and control of effective density / fill-in cost so that practical bandwidth savings can still be realized even in low-sparsity settings.
The experimental results are convincing. At 50% sparsity, across RTX 2080, 3090, and 4090, DELTA4 achieves clear memory savings and speedups relative to dense baselines, while also substantially outperforming sparse SpMV baselines such as cuSPARSE, Sputnik, and DASP. The paper further demonstrates the applicability of the method to pruned LLM inference, showing that under FP16 decoding workloads, Wanda-pruned models can achieve meaningfully lower memory usage and faster decoding while preserving a practical quality-speed trade-off. The discussion also strengthens the paper by clarifying that the effective density under real pruning patterns remains close to the expected case, that the current implementation still yields gains on H100 despite lacking Hopper-specific optimization, and that the method retains some benefit under INT8, while the gains under INT4 are inherently constrained by more fundamental information-theoretic limits.

Overall, this is a practically important paper that tackles sparse computation in the moderate unstructured sparsity regime, a setting of clear relevance to LLMs pruning. The work presents a thoughtful system-algorithm co-design, and the resulting gains in memory and latency are both meaningful and well supported by experiments. It also provides insights that could be valuable to the broader systems and efficient-LLM community. In addition, all reviewers recommend acceptance, which further supports a positive recommendation.